# Controlling Nutritional Status (CONUT) Score and Sarcopenia as Mutually Independent Prognostic Biomarkers in Advanced Urothelial Carcinoma

**DOI:** 10.3390/cancers14205075

**Published:** 2022-10-17

**Authors:** Minami Une, Masaya Ito, Hiroaki Suzuki, Masahiro Toide, Shuichiro Kobayashi, Hiroshi Fukushima, Fumitaka Koga

**Affiliations:** Department of Urology, Tokyo Metropolitan Cancer and Infectious Diseases Center Komagome Hospital, Tokyo 113-8677, Japan

**Keywords:** advanced urothelial carcinoma, controlling nutritional status score, sarcopenia, performance status, prognosis

## Abstract

**Simple Summary:**

While the controlling nutritional status (CONUT) score and sarcopenia are both objective indices of different aspects of a patient’s general condition, few studies have comprehensively examined their mutual relationship as prognostic factors. In the present, retrospective study, we examined this question using a cohort of 200 patients with advanced urothelial carcinoma (aUC). No significant association was found between the CONUT score and sarcopenia, and most patients with sarcopenic aUC had normal or only slightly impaired nutritional status. The CONUT score and sarcopenia were significant, mutually independent, prognostic biomarkers and they outperformed performance the status as a prognostic factor in our cohort. Incorporating the CONUT score, sarcopenia or both into current established prognostic models increased their predictive accuracy. Our study corroborated the prognostic relevance of the CONUT score and sarcopenia and suggested the importance of separately evaluating these prognostic biomarkers in patients with aUC.

**Abstract:**

Background: While the controlling nutritional status (CONUT) score and sarcopenia are objective indices of different aspects of a patient’s general condition, few studies have comprehensively examined their mutual relationship in patients with advanced cancer. Methods: This retrospective study included 200 Japanese patients with advanced urothelial carcinoma (aUC). Sarcopenia was diagnosed using Prado’s definition. The CONUT score and sarcopenia were examined for their possible association, and their prognostic value was analyzed. Results: The CONUT score and sarcopenia were not significantly associated. While sarcopenia occurred in 168 patients (84%), more than half of them had normal or only slightly impaired nutritional status, as indicated by a CONUT score of 0–2. During follow-up (median: 13.3 months), 149 patients died. The CONUT score and sarcopenia were independent prognostic factors (hazard ratio 1.22 and 2.23, respectively; both *p* < 0.001), whereas performance status was not. Incorporating the CONUT score, sarcopenia, and both into Bajorin’s and Apolo’s prognostic models increased their concordance index as follows: 0.612 for Bajorin’s original model to 0.653 (+the CONUT score), 0.631 (+sarcopenia), and 0.665 (+both), and 0.634 for Apolo’s original model to 0.655 (+the CONUT score), 0.653 (+ sarcopenia), and 0.668 (+both). Conclusion: The CONUT score and sarcopenia were mutually independent in terms of their prognostic value in patients with aUC. These objective indices of a patient’s general condition may help in decision-making when considering treatment for patients with aUC.

## 1. Introduction

Urothelial carcinoma (UC) is mainly comprised of bladder UC and upper tract (UT) UC (UTUC), and account for 90–95% and 5–10% of all UC cases, respectively [1]. Despite the recent advent of novel treatments, such as immune checkpoint inhibitors (ICIs) and enfortumab-vedotin, the prognosis of patients with advanced urothelial carcinoma (aUC) remains poor. Platinum-based regimens have been the standard of care for aUC since the late 1980s [2]. The median overall survival (OS) remained around 15 months, despite the efficacy of platinum-based chemotherapy [3,4,5,6]. While pembrolizumab and enfortumab-vedotin significantly prolonged the median OS, by approximately three months, as compared to second- and third-line chemotherapy after failure of first-line chemotherapy and second-line ICIs [7,8], respectively, only a small proportion of patients were able to extend their survival significantly. While several prognostic models of aUC have been proposed and externally validated [9,10], novel prognostic factors capable of better predicting the prognosis of aUC would help patients and their physicians to choose the optimal therapy.

Assessment of the general condition of a patient with advanced cancer is also crucial for decision-making and prognostication. Performance status (PS) is commonly used for these purposes. Despite its simplicity, the PS is a subjective assessment and thus includes possible problems with objectivity, accuracy, and interobserver variation [11]. Therefore, more objective and reliable tools for assessing the general condition of patients with advanced cancer are needed. The controlling nutritional status (CONUT) score [12] and sarcopenia may serve as such assessment tools from the perspective of nutritional status and body composition, respectively.

The CONUT score, based on the serum albumin level, lymphocyte count, and total cholesterol level (Table 1), is a simple, validated, objective tool for assessing nutritional status [12,13]. The CONUT score has been used as a prognostic factor in patients with various malignancies [14,15,16,17], including aUC [18,19].

Sarcopenia is defined as a decrease in skeletal muscle mass and function [20]. Most of the numerous studies examining sarcopenia [21,22,23,24,25] used the skeletal muscle index (SMI), based on computed tomography (CT) images. As with the CONUT score, sarcopenia has been used as a prognostic factor in a variety of cancers, including hepatocellular carcinoma, lung cancer, breast cancer, and aUC [26,27,28,29]. Moreover, sarcopenia is associated with an increased risk of adverse events related to chemotherapy and major surgery [30,31,32,33].

While both the CONUT score and sarcopenia reflect the deterioration in patients’ general condition associated with tumor-induced, chronic inflammation and cachexia and can provide prognostic information independently of canonical, prognostic factors in patients with various malignancies, no study has comprehensively investigated whether the CONUT score and sarcopenia are associated or are mutually independent prognostic factors. In the present study, we assessed the prognostic value of the CONUT score and sarcopenia in 200 patients with aUC, most of whom had received platinum-based systemic chemotherapy.

## 2. Materials and Methods

### 2.1. Study Design, Data Collection, and CONUT Score Calculation

Our institutional ethics committee approved the present retrospective study (approval number: 2894). In total, 247 consecutive Japanese patients with inoperable (cT4 or lymph node metastasis) and/or metastatic UC of the bladder or UT were treated at a single, designated cancer center between December 2002 and December 2021. Of the 247 patients, 47 were excluded owing to missing data, which was required to calculate the CONUT score (n = 30) or missing CT imaging studies (n = 17). Finally, 200 patients were enrolled. Data at the diagnosis of aUC on age, sex, Eastern Cooperative Oncology Group (ECOG) PS, body mass index (BMI), the primary tumor site (bladder or UT), hydronephrosis, lymph node or visceral metastasis, curative surgery before and after the aUC diagnosis, first line therapy for aUC, hemoglobin, neutrophil and lymphocyte counts, creatinine, albumin, alkaline phosphatase (ALP), lactate dehydrogenase (LDH), corrected calcium, C-reactive protein (CRP), total cholesterol, SMI, the CONUT score, and sarcopenia were collected from the medical records. The CONUT score was calculated using values for serum albumin, total lymphocyte count, and total cholesterol concentration (Table 1) [12]. BMI was calculated using the formula: BMI (kg/m^2^) = ((weight)/(height)^2^). The first line treatment of aUC was classified into: (1) Platinum-based chemotherapy; (2) chemoradiotherapy with a curative intent; and (3) best supportive care (BSC). Chemoradiotherapy was mainly offered to patients with locally advanced disease who were unfit for systemic chemotherapy. Low-dose cisplatin or fluorouracil was given concurrently with chemoradiotherapy as a radiosensitizer.

The primary aim of this study was to assess whether the CONUT score and sarcopenia are mutually independent, prognostic biomarkers in aUC.

### 2.2. Image Analysis Evaluating Sarcopenia

CT was performed for diagnosis or follow-up. Axial CT images taken within 30 days of diagnosis of aUC were used. The third lumbar vertebra (L3) was chosen as a landmark, and one slice was selected to measure the cross-sectional area of skeletal muscle using Hounsfield unit thresholds of −29 to +150 [21,34]. Skeletal muscle at the L3 level included the psoas, paraspinal muscles (erector spinae and quadratus lumborum), and abdominal wall muscles (transversus abdominus, external and internal obliques, and rectus abdominus). The total, cross-sectional area of lumbar skeletal muscle was linearly related to the cross-sectional area of the whole-body muscle [35]. To assess for sarcopenia, the total muscle area was normalized for stature, as is usually done for BMI and body composition assessments using the formula: Skeletal muscle index (SMI) (cm^2^/m^2^) = ((skeletal muscle cross-sectional area at L3)/(height)^2^) [21,34]. Images were analyzed using OsiriX Lite (Pixmeo, Geneva, Switzerland; https://www.osirix-viewer.com/ (accessed on 1 July 2022.)). Image analysis was performed by three investigators (M.U., H.S. and H.F.), who were blinded to other variables and patient outcomes. The representative definitions used to diagnose sarcopenia based on SMI included Prado’s definition (SMI < 52.4 cm^2^/m^2^ for male patients and <38.5 cm^2^/m^2^ for female patients) Martin’s definition (SMI < 43 cm^2^/m^2^ for male patients with a BMI < 25; SMI < 53 cm^2^/m^2^ for male patients with BMI ≥ 25; and SMI < 41 cm^2^/m^2^ for female patients), and the international definition (SMI < 55 cm^2^ /m^2^ for male patients and <39 cm^2^/m^2^ for female patients) (14–16). The prognostic value of sarcopenia was assessed according to each definition, and the definition in which sarcopenia was found to have the greatest prognostic value (the lowest *p*-value and highest Harrell’s concordance index [c-index]) was adopted.

### 2.3. Statistical Analysis

Differences in the distribution of the variables between the groups were evaluated using Fisher’s exact test for categorical variables and the Wilcoxon rank sum test for continuous variables. OS was defined as the time from aUC diagnosis to either death or the last follow-up (data cutoff date: 31 December 2021). Martingale residuals were plotted for the CONUT score to judge the goodness of linear fit of the prognostic effects [29]. Patients were divided into three groups using two CONUT score cut-off values showing the greatest between-group differences in the OS curves. The survival curves were estimated using the Kaplan-Meier method, and differences between the groups were evaluated using the log-rank test. Univariate and multivariate Cox proportional hazards were used to test for any association between the variables and OS. The c-index was used to estimate the predictive accuracy of the prognostic models. To validate Apolo’s and Bajorin’s models, both of which include Karnofsky PS (K-PS) < 80% as one of the variables [9], K-PS < 80% was substituted with ECOG PS < 2 [36]. All statistical analyses were conducted using JMP 14.0.0. (SAS Institute Inc., Cary, NC, USA) and R 4.1.0 (R Foundation for Statistical Computing, Vienna, Austria). *p* < 0.05 was considered to indicate statistical significance.

## 3. Results

### 3.1. Patient Characteristics

Table 2 shows the demographic data of all 200 patients with aUC. The median (range) age at diagnosis was 71 (38–94) years, and the primary tumor site was the bladder and UT in 109 (55%) and 91 patients (45%), respectively. The ECOG PS was 0, 1, ≥2 in 136 (68%), 42 (21%), and 22 patients (11%), respectively. Forty-seven patients (23%) had previously undergone curative surgery. Metastases to lymph nodes and visceral organs and unresectable T4 disease were observed in 82 (41%), 78 (39%), and 44 patients (22%), respectively. In total, 163 (82%) patients received first-line therapy, such as platinum-based systemic chemotherapy (n = 148, 74%; n = 117, 59% for cisplatin and n = 31, 16% for carboplatin) or chemoradiation (n = 15, 8%). Meanwhile, 37 (19%) patients were placed on BSC. Pembrolizumab was given as a second or later line systemic therapy in 32 patients (16%), and 34 patients (17%) underwent curative surgery after achieving an objective response to systemic chemotherapy (n = 31, 16%) or chemoradiation (n = 3, 2%).

The CONUT score was 0, 1, 2, 3, and ≥4 in 37 (19%), 66 (33%), 43 (22%), 29 (14%), and 25 patients (12%), respectively, and was subgrouped into scores 0–1 (n = 103, 52%), 2–3 (n = 72, 36%), and ≥4 (n = 25, 12%) according to its association with OS. A higher CONUT score was significantly associated with a poorer ECOG PS (*p* < 0.001), lower BMI (*p* < 0.001), presence of hydronephrosis (*p* < 0.001), higher proportion of BSC (*p* = 0.044), lower hemoglobin (*p* < 0.001), higher neutrophil count (*p* = 0.011), higher ALP (*p* = 0.006), higher CRP (*p* < 0.001), lower SMI (*p* = 0.017), as well as lower albumin, lymphocyte count, and total cholesterol (all *p* < 0.001).

Table 3 shows that Prado’s definition had the highest prognostic value of the three definitions of sarcopenia. Based on Pardo’s definition, sarcopenia was observed in 168 patients (84%) and was more prevalent in male patients (*p* = 0.012) and patients with primary bladder UC (*p* = 0.019), in comparison to their counterparts, and was significantly associated with a lower BMI (*p* < 0.001), more frequent carboplatin use (*p* = 0.009), lower lymphocyte count (*p* = 0.030), and lower total cholesterol (*p* = 0.039, Table 2). No significant association was observed between sarcopenia and the CONUT score (*p* = 0.219); the median (range) CONUT score was 2 (0–8) and 1 (0–7) in patients with and without sarcopenia, respectively (*p* = 0.416).

### 3.2. Association of the CONUT Score and Sarcopenia with OS

During follow-up (median: 13.3 months; range 1–183 months), 149 patients died. Figure 1 shows that the OS curves differed significantly among the CONUT scores 0–1, 2–3, and ≥4 with a median OS of 19.7 months, 14.5 months and 9.1 months, respectively (*p* < 0.001). Figure 2 shows that patients with sarcopenia had significantly shorter OS than those without sarcopenia (median OS: 14.5 vs. 20.9 months; *p* = 0.027).

Table 4 shows the association of the variables with OS in univariable and multivariable analyses. In the univariable analysis, the CONUT score and sarcopenia were significantly associated with poor OS (hazard ratio [HR]: 1.18, *p* = 0.036; and HR: 2.28; *p* = 0.006, respectively). The following variables were also significantly associated with poor OS: age (HR: 1.03; *p* = 0.013), presence of hydronephrosis (HR: 1.50; *p* = 0.049), the UT primary tumor (HR: 1.90; *p* = 0.034), visceral metastasis (HR: 1.56; *p* = 0.034), previous curative surgery (HR: 1.11; *p* = 0.002), high neutrophil count (HR: 1.01; *p* = 0.013), high ALP (HR: 1.02; *p* < 0.001), and high LDH (HR: 1.01; *p* = 0.003). Multivariable analysis demonstrated that a high CONUT score (HR: 1.22; *p* < 0.001) and sarcopenia (HR: 2.23; *p* < 0.001), along with the following variables, were independently associated with poor OS: age (HR: 1.04; *p* = 0.002), presence of hydronephrosis (HR: 1.53; *p* = 0.016), UT primary tumor (HR: 1.69; *p* = 0.004), visceral metastasis (HR: 1.71; *p* = 0.004), high neutrophil count (HR: 1.01; *p* = 0.005), high ALP (HR: 1.02; *p* < 0.001), high LDH (HR: 1.01; *p* = 0.003), and high CRP (HR: 1.01; *p* = 0.033).

To confirm the prognostic independence of the CONUT score and sarcopenia, a sensitivity analysis was conducted. For 168 patients with sarcopenia (Appendix A), the CONUT score was significantly associated with OS (univariable HR: 1.35; 95% CI: 1.20–1.51; *p* < 0.001). While the OS curves were similarly separated (Appendix A) in 32 patients without sarcopenia, statistical significance was not observed, probably due to the small sample size (univariable HR: 1.16; 95% CI: 0.92–1.46; *p* = 0.218). In 175 patients with the CONUT score 0–3, sarcopenia was significantly associated with shorter OS (*p* = 0.039, Appendix A). OS curves were similarly separated in 25 patients with the CONUT score ≥4, but no statistical difference was observed in this small subgroup (*p* = 0.394, Appendix A).

### 3.3. Role of the CONUT Score and Sarcopenia as Prognostic Factors

Next, we evaluated the role of the CONUT score and sarcopenia as prognostic factors in a subgroup of 148 patients who had received first-line platinum-based systemic chemotherapy using two established prognostic models for such patients. When adding the CONUT score instead of Alb, sarcopenia, and both of Apolo’s models, the c-index increased from 0.634 to 0.655, 0.653, and 0.668, respectively. Similarly, the c-index increased from 0.612 to 0.653, 0.631 and 0.665, respectively, with Bajorin’s model (Table 5).

## 4. Discussion

Both the CONUT score and sarcopenia are known to be objective indicators of a patient’s general condition and to provide prognostic information about patients with various malignancies. However, no study has examined whether there is any association between the two indicators or what their comprehensive prognostic role in patients with advanced cancer might be. To the best of our knowledge, the present study is the first to address these questions. First, our study found no significant association between the CONUT score and sarcopenia diagnosed using Prado’s definition in our cohort of 200 patients with aUC. Second, both the CONUT score and sarcopenia were independent prognostic factors of OS, and these indices were mutually independent in terms of their prognostic value. In fact, adding the CONUT score, sarcopenia, and both to Apolo’s and Bajorin’s prognostic models, improved the models’ predictive accuracy in 148 patients who had received first-line platinum-based chemotherapy. Our findings suggested that the CONUT score and sarcopenia, which are objective indices of a patient’s general condition in terms of nutritional status and body composition, respectively, are independently associated with the prognosis of patients with aUC. This study corroborated the prognostic value of a patient’s general condition and indicated the importance of separately evaluating the CONUT score and sarcopenia as prognostic factors in aUC.

PS is an established prognostic factor of aUC [9,10]. However, it demonstrated no prognostic value in the present study, possibly because the subjectivity of PS led to inaccuracies in its assessment by multiple physicians [37]. At any rate, the CONUT score and sarcopenia outperformed PS as prognostic factors in the present study.

According to the international consensus, cancer cachexia is defined as a multifactorial, complex, metabolic syndrome characterized by an ongoing loss of skeletal muscle mass that cannot be fully reversed by conventional nutritional support [22]. In patients with cancer cachexia, the metabolic balance shifts towards catabolism rather than anabolism, owing mainly to cancer-related systemic inflammation and anorexia, which lead to sarcopenia and malnutrition [38,39]. Thus, sarcopenia and the CONUT score can reflect the presence and severity of cancer cachexia, respectively. In the present study, the CONUT score was inversely associated with SMI, while sarcopenia was associated with lower lymphocyte counts and cholesterol levels. However, no significant association was observed between the CONUT score and sarcopenia. The lack of the association could be the consequence of the chosen cut-off of SMI according to the Prado’s definition. While sarcopenia was observed in 84% of aUC patients, more than half of the sarcopenic patients had normal or only slightly impaired nutritional status, as indicated by a CONUT score of 0–2, thus suggesting that malnutrition, particularly hypoalbuminemia, develops at a later phase of cachexia than sarcopenia in the majority of aUC patients. It is notable that more than 20% of patients with aUC without malnutrition (CONUT score 0–1) and those without sarcopenia survived five years or longer (Figure 1 and Figure 2). As less cachexic patients are also more tolerant of intensive therapy [40,41], they may be good candidates for intensive aUC treatment.

Currently, the optimal definition of SMI-based sarcopenia has not been standardized. The present study compared the prognostic value of three major definitions of sarcopenia [21,22,23] and demonstrated that Prado’s definition most significantly stratified the prognosis (Table 3). The three definitions of sarcopenia originated from the same Canadian cohort. Given that the Japanese population generally have smaller body frame than their Canadian counterparts, Prado’s definition with its lower SMI thresholds may more accurately reflect sarcopenia in Japanese patients than the international definition. Indeed, a multiracial cohort study of healthy elderly people demonstrated that the prevalence of sarcopenia was higher in Asians than other racial groups including Whites, Blacks, and Hispanics when a common definition of sarcopenia was applied [42]. Some researchers have questioned the scientific basis of Martin’s definition, in which the SMI threshold for men depends on BMI and is therefore discontinuous [43]. In fact, a preliminary study of 64 Japanese patients with aUC demonstrated that, of the three major definitions, Prado’s definition was the best predictor of cancer-specific survival, while Martin’s definition failed to demonstrate a significant difference between patients with and without sarcopenia [44]. Given that body frame varies among racial groups, different definitions of SMI-based sarcopenia may be needed depending on racial groups.

The present study had several limitations. First, its retrospective, monocentric design may have introduced a bias. External validation using a large, multicentric cohort is needed to verify the generalizability of the mutual independence of the CONUT score and sarcopenia in aUC patients. Second, about 20% (47/250) of patients were excluded because some of the data pertaining to the CONUT sore (mainly total cholesterol) or sarcopenia in the initial cohort were missing. There may also be some bias associated with this exclusion. Third, use of statins, which influence the total cholesterol level, was not included as a variable because the relevant data were incomplete. Fourth, only 32 patients received pembrolizumab, the current, standard, second-line treatment for aUC and none received maintenance avelumab following platinum-based chemotherapy. As pembrolizumab and avelumab were approved in December 2017 and February 2021, respectively, in Japan, most of the patients were unable to receive ICIs. Different results might be obtainable in a new cohort receiving ICIs. Fifth, our study cohort consisted of only Japanese patients and thus further studies on other racial groups are needed to validate the generality of our findings.

## 5. Conclusions

The present study demonstrated the prognostic significance of the CONUT score and sarcopenia, which were found to be mutually independent in Japanese patients with aUC. These objective indices of a patient’s general condition in terms of nutritional status and body composition, respectively, may help in decision-making when choosing a treatment for aUC.

## Figures and Tables

**Figure 1 cancers-14-05075-f001:**
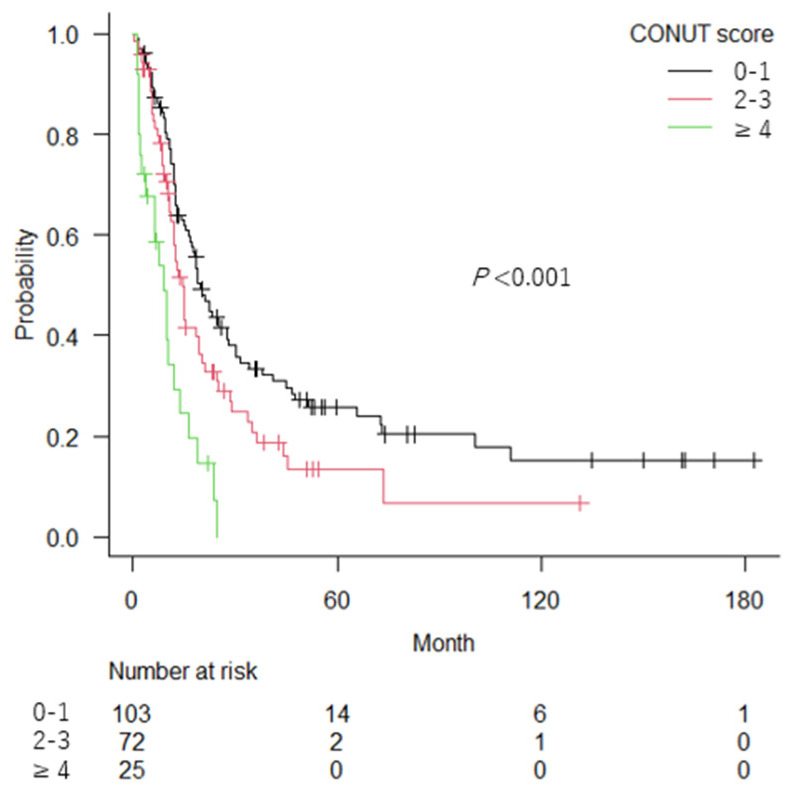
Kaplan–Meier curves for overall survival in patients with advanced urothelial carcinoma according to the controlling nutritional status (CONUT) score.

**Figure 2 cancers-14-05075-f002:**
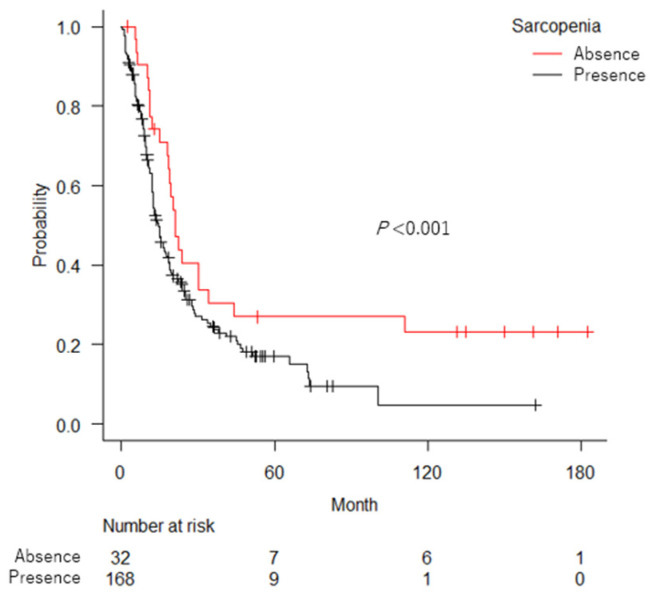
Kaplan–Meier curves of overall survival in patients with advanced urothelial carcinoma according to the presence or absence of sarcopenia.

**Table 1 cancers-14-05075-t001:** Scoring and interpretation of the controlling nutritional status (CONUT) score.

Parameter	Range of Values and Scores Per Parameter			
Albumin (g/dL)	≥3.50	3.00–3.49	2.50–2.99	<2.50
Score	0	2	4	6
Lymphocyte count (/μL)	≥1600	1200–1599	800–1199	<800
Score	0	1	2	3
Total cholesterol (mg/dL) *	≥180	140–179	100–139	<100
Score	0	1	2	3
Interpretation				
CONUT score (sum of above scores)	0–1	2–4	5–8	9–12
Degree of malnutrition	None	Light	Moderate	Severe

* Ranges of total cholesterol (mmol/L) are ≥4.65, 3.62–4.64, 2.58–3.61, and <2.58, respectively.

**Table 2 cancers-14-05075-t002:** Baseline characteristics of 200 patients with advanced urothelial carcinoma.

Variable		Total, N (%)	CONUT Score, N (%)			*p*	Sarcopenia, N (%)		*p*
			0–1	2–3	≥4		Yes	No	
Total		200 (100)	103 (52)	72 (36)	25 (12)		168 (84)	32 (16)	
Age (years) *		71 (38–94)	70 (38–86)	72 (46–90)	72 (57–94)	0.334	71 (38–91)	69 (47–94)	0.188
Sex						0.426			0.012
Male		138 (69)	70 (68)	53 (74)	15 (60)		122 (73)	16 (50)	
Female		62 (31)	33 (32)	19 (26)	10 (40)		46 (27)	16 (50)	
ECOG PS						<0.001			0.141
0		136 (68)	73 (71)	54 (75)	9 (36)		115 (68)	21 (66)	
1		42 (21)	24 (23)	12 (17)	6 (24)		32 (19)	10 (31)	
≥2		22 (11)	6 (6)	6 (8)	10 (40)		21 (13)	1 (3)	
BMI (kg/m^2^) *		22 (15–36)	23 (15–36)	22 (15–32)	21 (17–27)	<0.001	22 (15–34)	24 (18–36)	<0.001
Primary tumor site						0.599			0.019
Bladder		109 (55)	58 (56)	36 (50)	15 (60)		98 (58)	11 (34)	
UT		91 (45)	45 (44)	36 (50)	10 (40)		70 (42)	21 (66)	
Hydronephrosis						<0.001			0.127
No		106 (53)	56 (54)	44 (61)	6 (24)		85 (51)	21 (66)	
Yes		94 (47)	47 (46)	28 (39)	19 (76)		83 (49)	11 (34)	
Lymph node metastasis						0.716			0.245
No		118 (59)	62 (60)	40 (56)	16 (64)		96 (57)	22 (69)	
Yes		82 (41)	41 (40)	32 (44)	9 (36)		72 (43)	10 (31)	
Visceral metastasis						0.072			0.845
No		122 (61)	67 (65)	45 (63)	10 (40)		103 (61)	19 (59)	
Yes		78 (39)	36 (35)	27 (38)	15 (60)		65 (39)	13 (41)	
Prior curative surgery						0.819			0.066
No		153 (77)	80 (78)	55 (76)	18 (72)		133 (79)	20 (63)	
Yes		47 (23)	23 (22)	17 (24)	7 (28)		35 (21)	12 (38)	
1st line therapy for aUC						0.044			0.546
Platinum-based chemotherapy	cisplatin	117 (59)	67 (34)	40 (20)	10 (5)	0.782	93(47)	24(12)	0.009
	carboplatin	31 (16)	17 (9)	10 (5)	4 (2)		30(15)	1(1)	
Chemoradiation		15 (7.5)	5 (33)	8 (54)	2 (13)		14 (93)	1 (7)	
BSC		37 (19)	14 (38)	14 (38)	9 (24)		30 (81)	7 (19)	
Curative surgery after diagnosis of aUC						0.117			0.774
No		166 (83)	80 (48)	64 (39)	22 (13)		140 (84)	26 (16)	
Yes		34 (17)	23 (68)	8 (24)	3 (8)		28 (82)	6 (18)	
Administration of pembrolizumab for 2nd or later line therapy		32 (16)	19 (59)	12 (38)	1 (3)	0.206	31 (18)	1 (3)	0.030
Hemoglobin (g/dL) *		12 (3.1–18)	13 (7.7–18)	11 (6.2–16)	11 (3.1–13)	<0.001	12 (3.1–18)	12 (6.2–15)	0.109
Neutrophil (×10^3^/μL) *		4.9 (1.1–56)	5.2 (1.1–31)	4.3 (1.8–12)	6.7 (1.7–56)	0.011	6.9 (1.0–59)	8.0 (4.2–18)	0.060
Lymphocyte (×10^3^/µL) *		1.5 (0.17–4.3)	1.8 (1.2–4.3)	1.1 (0.64–2.7)	1.1 (0.17–2.3)	<0.001	1.4 (0.17–4.0)	1.8 (0.52–4.3)	0.030
Creatinine (mg/dL) *		1.0 (0.49–16)	1.1 (0.49–4.3)	1.0 (0.50–3.1)	1.1 (0.50–16)	0.298	1.0 (0.50–16)	1.0 (0.50–3.1)	0.450
Albumin (g/dL) *		4.0 (2.7–5.0)	4.1 (3.6–5.0)	4.0 (3.2–4.9)	3.2 (2.7–4.2)	<0.001	4 (2.7–5.0)	4 (2.8–4.7)	0.979
ALP (U/L) *		261 (92–3351)	253 (92–1539)	258 (103–3351)	307 (212–682)	0.006	266 (92–3351)	237 (147–586)	0.142
LDH (U/L) *		191 (118–2970)	191 (119–2482)	191 (118–2970)	212 (129–386)	0.382	192 (129–2970)	187 (118–880)	0.288
Corrected calcium (mg/dL) *		8.8 (7.6–14)	8.8 (7.6–14)	8.7 (7.9–11)	8.8 (7.6–11)	0.829	8.8 (7.6–14)	8.7 (7.9–9.3)	0.168
CRP (mg/L) *		6.0 (0.00–266)	3.0 (0.40–115)	5.7 (0.00–266)	34 (0.60–139)	<0.001	6.4 (0.0–266)	6.0 (0.60–65)	0.632
Total cholesterol (mg/dL) *		183 (98–275)	195 (140–271)	175 (98–275)	167 (118–224)	<0.001	169 (126–240)	187 (98–275)	0.039
SMI (cm^2^/m^2^) *		37 (16–64)	38 (21–64)	36 (16–62)	33 (18–54)	0.017	35 (16–51)	53 (39–64)	<0.001
CONUT score *		2 (0–8)	1 (0–1)	2 (2–3)	5 (4–8)	-	2 (0–8)	1 (0–7)	0.416
Sarcopenia						0.219			-
Yes		168 (84)	82 (80)	64 (89)	22 (88)		168 (100)	0 (0)	
No		32 (16)	21 (20)	8 (11)	3 (12)		0 (0)	32 (100)	

CONUT, controlling nutritional status; ECOG PS, Eastern Cooperative Oncology Group performance status; BMI, body mass index; UT, upper tract; UC, urothelial carcinoma; BSC, best supportive care; ALP, alkaline phosphatase; LDH, lactate dehydrogenase; CRP, C-reactive protein; SMI, skeletal mass index. * Median (range).

**Table 3 cancers-14-05075-t003:** Median overall survival and concordance index for each definition of Sarcopenia.

Definition of Sarcopenia	Sarcopenia	Median OS (Range, Months)	*p*	C-Index
Prado’s definition	Absent	20.5 (2–183)	0.003	0.541
	Present	12.5 (1–162)		
Martin’s definition	Absent	18.7 (1–183)	0.214	0.516
	Present	12.7 (1–162)		
International definition	Absent	20.9 (2–183)	0.020	0.530
	Present	12.7 (1–162)		

OS, overall survival; c-index, concordance-index.

**Table 4 cancers-14-05075-t004:** Univariable and multivariable analyses of overall survival in 200 patients with advanced urothelial carcinoma.

Variables	Univariable			Multivariable (Final Model)		
	HR	(95% CI)	*p*	HR	(95% CI)	*p*
Age	1.03	(1.01–1.06)	0.013	1.04	(1.02–1.06)	0.002
Sex						
Female (vs. male)	1.38	(0.89–2.13)	0.134			
ECOG PS						
0	ref					
1	1.39	(0.85–2.28)	0.184			
≥2	1.69	(0.85–3.34)	0.133			
BMI	0.97	(0.92–1.02)	0.170			
Hydronephrosis						
Yes (vs. no)	1.50	(1.00–2.26)	0.049	1.53	(1.08–2.17)	0.016
Primary site						
UT (vs. bladder)	1.90	(1.26–2.85)	0.034	1.69	(1.18–2.41)	0.004
Lymph node metastasis						
Yes (vs. no)	0.94	(0.64–1.37)	0.758			
Visceral metastasis						
Yes (vs. no)	1.56	(1.03–2.33)	0.034	1.71	(1.18–2.44)	0.004
Previous curative surgery						
Yes (vs. no)	1.11	(1.26–2.85)	0.002			
1st line therapy for aUC						
Chemotherapy	ref					
CRT	1.32	(0.61–2.67)	0.452			
BSC	1.05	(0.59–1.82)	0.862			
Curative surgery after diagnosis of aUC						
Yes (vs. no)	0.64	(0.36–1.10)	0.108			
Hemoglobin	1.09	(0.96–1.01)	0.203			
Neutrophil †	1.01	(1.00–1.01)	0.013	1.01	(1.00–1.01)	0.005
Lymphocyte †	1.00	(0.96–1.03)	0.900	-	-	-
Creatinine	1.13	(0.97–1.29)	0.103			
Albumin	0.79	(0.42–1.47)	0.460	-	-	-
ALP ††	1.02	(1.01–1.03)	<0.001	1.02	(1.01–1.03)	<0.001
LDH ††	1.01	(1.01–1.02)	0.003	1.01	(1.01–1.02)	0.003
Corrected calcium	1.24	(0.88–1.66)	0.212			
CRP	1.13	(0.97–1.29)	0.066	1.01	(1.00–1.01)	0.033
Total cholesterol	1.00	(0.99–1.01)	0.997	-	-	-
Sarcopenia						
Yes (vs. no)	2.28	(1.26–4.27)	0.006	2.23	(1.39–3.72)	<0.001
CONUT score	1.18	(1.01–1.37)	0.036	1.22	(1.09–1.36)	<0.001

HR, hazard ratio; CI, confidence interval; ECOG PS, Eastern Cooperative Oncology Group performance status; BMI, body mass index; UT, upper tract; UC, urothelial carcinoma; ALP, alkaline phosphatase; LDH, lactate dehydrogenase; CRP, C-reactive protein; CONUT, controlling nutritional status; NS, not significant. † every 100 units; †† every 10 units.

**Table 5 cancers-14-05075-t005:** C-index of the established prognostic models combined with the controlling nutritional status (CONUT) score or/and sarcopenia.

Prognostic Model	C-Index	Prognostic Model	C-Index
Apolo’s model (KPS + visceral metastasis + Hb + Alb)	0.634	Bajorin’s model (KPS + visceral metastasis)	0.612
Apolo’s model + the CONUT score—Alb	0.655	Bajorin’s model + the CONUT score	0.653
Apolo’s model + sarcopenia	0.653	Bajorin’s model + sarcopenia	0.631
Apolo’s model + the CONUT score—Alb + sarcopenia	0.668	Bajorin’s model + the CONUT score + sarcopenia	0.665

KPS, Karnofsky performance status; Hb, hemoglobin; Alb, albumin; the CONUT score, controlling nutritional status score.

## Data Availability

The data can be shared up on request.

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
