# Peer review of "Controlling Nutritional Status (CONUT) Score and Sarcopenia as Mutually Independent Prognostic Biomarkers in Advanced Urothelial Carcinoma"

_cancers, 2022, doi:10.3390/cancers14205075_

Round 1
Reviewer 1 Report
Congratulations on the paper, the article is well-written and provides important information on the prognostic value of nutritional status on survival in aUC.
1. You claimed that “No significant association was observed between sarcopenia 174 and the CONUT score (P = 0.219)” – how was this relationship tested? Could you provide a contingency table showing the relationship between sarcopenia and CONUT score?
In the table 2 you show that SMI (skeletal mass index differs between different CONUT points). Considering that the lack of significant association between CONUT and sarcopenia might be just the consequence of the chosen cutoff of SMI for sarcopenia diagnosis. Nevertheless I still believe that both CONUT and sarcopenia provide important and separate prognostic value. But please comment on that issue.
2. Did you try to use ECOG (or Karnofsky) as three-level ordinal variable in univariate and multivariate analysis? You simply categorized ECOG and showed that it is not significant. Please try to use ECOG as ordinal variable (such as you presented in Table 2) in Cox proportional hazard analysis.
3. How was the C-index calculated in the table 5? Was it derived from Cox proportional hazards? If yes which time-point was chosen?
Thank you
Author Response
First of all, we greatly appreciated the reviewers for taking their precious time and giving us constructive comments. As listed below, we replied to their comments in a point-by-point manner. We hope the revised version is suitable for publication in Cancers.
Reviewer 1
Congratulations on the paper, the article is well-written and provides important information on the prognostic value of nutritional status on survival in aUC.
We appreciate the reviewer’s favorable general comments.
- You claimed that “No significant association was observed between sarcopenia 174 and the CONUT score (P = 0.219)” – how was this relationship tested? Could you provide a contingency table showing the relationship between sarcopenia and CONUT score?
In the table 2 you show that SMI (skeletal mass index differs between different CONUT points). Considering that the lack of significant association between CONUT and sarcopenia might be just the consequence of the chosen cutoff of SMI for sarcopenia diagnosis. Nevertheless I still believe that both CONUT and sarcopenia provide important and separate prognostic value. But please comment on that issue.
Reply:
We appreciate the reviewer’s important comments. We tested the association between sarcopenia and CONUT scores using a contingency table (shown in bottoms of Table 2, p = 0.219) and Wilcoxon rank sum test (the line of “CONUT score” in Table 2, p = 0.416). As the reviewer pointed out, SMI was inversely associated with CONUT scores (Table 2, p = 0.017). We totally agree to the reviewer’s opinion that the lack of significant association between CONUT and sarcopenia could be just the consequence of the chosen cutoff of SMI for sarcopenia diagnosis. In the revised version, we added the following sentence in the 3rd paragraph of the Discussion section; “The lack of the association could be the consequence of the chosen cut-off of SMI according to the Prado’s definition.”
- Did you try to use ECOG (or Karnofsky) as three-level ordinal variable in univariate and multivariate analysis? You simply categorized ECOG and showed that it is not significant. Please try to use ECOG as ordinal variable (such as you presented in Table 2) in Cox proportional hazard analysis.
Reply:
We revised Table 4 by replacing dichotomized ECOG PS with PS 0, 1, vs. 2 or higher as presented in Table 2. As shown in revised Table 4, ECOG PS did not show prognostic significance in our cohort. 
- How was the C-index calculated in the table 5? Was it derived from Cox proportional hazards? If yes which time-point was chosen?
Reply:
We calculated c-indices using the “coxph” function of survival package in R software. In this software, c-index was automatically calculated when the Cox proportional hazard model was run. We do not need to choose a certain time point for outcome variables (dead or alive) because the Cox analysis per se deals with time-dependent outcome variables. As this software is commonly used for calculating c-index, we believe our analyses are methodologically sound.

Reviewer 2 Report
Generally taken well written paper. However, some minor corrections are suggested:
Line 32: please rewrite the sentence, because it is hard to understand at the moment. For example first numbers for Bajorin .... and then for Apolo ....
In table 1: please provide also European values (mmol) for cholesterol.
Table 2: Please write in Total also sign of percent: Total (%), and write range when appropriate, for example Age in years (range).
In general: Did you record cisplatin vs. carboplatin and were there any difference among these patients? If not done previously, is this analysis possible to do afterwards?
Discussion: How these results can be generalized, if the the best sarcopenia index is working maybe only in Japanese patients? Please discuss more based on the literature.
Also other immuno-oncological treatments are accepted to be used in the second line. Any of those? Please mention this.
In conclusion: In my opinion, the sentence too wide, as this is tested now only in Japanese patients. Please rewrite.
Author Response
First of all, we greatly appreciated the reviewers for taking their precious time and giving us constructive comments. As listed below, we replied to their comments in a point-by-point manner. We hope the revised version is suitable for publication in Cancers.
Reviewer 2
Generally taken well written paper. However, some minor corrections are suggested:
We appreciate the reviewer’s favorable general comments.
- Line 32: please rewrite the sentence, because it is hard to understand at the moment. For example first numbers for Bajorin .... and then for Apolo ....
Reply:
We agree with the Reviewer 2. We revised that part as following: “Incorporating the CONUT score, sarcopenia, and both into Bajorin’s and Apolo’s prognostic model increased their concordance index as follows: 0.612 for Bajorin’s original model to 0.653 (+ the CONUT score), 0.631 (+ sarcopenia), and 0.665 (+ both), and 0.634 for Apolo’s original model to 0.655 (+ the CONUT score), 0.653 (+ sarcopenia), and 0.668 (+ both).”
- In table 1: please provide also European values (mmol) for cholesterol.
Reply:
Ranges of total cholesterol in European values (mmol/L) are listed in the footnote of Table 1 as follows. “*Ranges of total cholesterol (mmol/L) are ≥4.65, 3.62-4.64, 2.58-3.61, and <2.58, respectively.”
- Table 2: Please write in Total also sign of percent: Total (%), and write range when appropriate, for example Age in years (range).
Reply:
In the revised version, “Total” was corrected as “Total, N (%)”. Variables shown in “median (range)” have been indicated by asterisks.
- In general: Did you record cisplatin vs. carboplatin and were there any difference among these patients? If not done previously, is this analysis possible to do afterwards?
Reply:
We appreciated the reviewer’s insightful suggestion. We did not find a significant association with the CONUT score (p=0.782 whereas did find more frequent carboplatin use in sarcopenic patients than those without sarcopenia (p=0.009). We added these findings in Table 2 and the Results section. These results suggest the presence of an association of chronic kidney disease with sarcopenia, a known fact [Wang et al., Nat Rev Nephrol 10:504-16, 2014], in aUC patients.
- Discussion:How these results can be generalized, if the the best sarcopenia index is working maybe only in Japanese patients? Please discuss more based on the literature.
Reply:
The current biggest problem for image-based diagnosis of sarcopenia is the lack of a standardized definition across racial groups. However, it would be inevitable that different SMI cut-offs are required depending on races when considering racial differences in body frame. In fact, multiracial studies on healthy elderly people reported that appendicular lean mass index was lower in Asians than other races (Whites, Blacks, and Latinos) [Wu et al., JCSM 13:987-1002, 2022 and Jeng et al. JCSM Clin Rep 3: e00027, 2018] and that the prevalence of sarcopenia was highest in Asians [Jeng et al. JCSM Clin Rep 3:e00027, 2018]. In the present study, we selected Prado’s definition among three major published definitions of sarcopenia because its prognostic value was most significant in our Japanese aUC patient cohort. Similarly, appropriate definitions of sarcopenia that discriminate prognosis may vary depending on racial groups of study cohorts. Further studies in other racial groups than Japanese are needed to assess the generality of our findings that the CONUT score and sarcopenia are mutually independent prognosticators in aUC patients.
We extended discussion on this topic according to the reviewer’s suggestion.
- Also, other immuno-oncological treatments are accepted to be used in the second line. Any of those? Please mention this.
Reply:
In Japan, 2nd line pembrolizumab and maintenance avelumab are available for aUC. They were approved in December 2017 and in February 2021, respectively, and our study cohort did not include patients receiving avelumab. We added information on avelumab in the Discussion section.
- In conclusion: In my opinion, the sentence too wide, as this is tested now only in Japanese patients. Please rewrite.
Reply:
We revised the Conclusions section as suggested by the reviewer.

Round 2
Reviewer 1 Report
No further comments, thank you